# Development of A Standardized Opsonophagocytosis Killing Assay for Group B *Streptococcus* and Assessment in an Interlaboratory Study

**DOI:** 10.3390/vaccines11111703

**Published:** 2023-11-09

**Authors:** Stephanie Leung, Clare F. Collett, Lauren Allen, Suzanna Lim, Pete Maniatis, Shanna J. Bolcen, Bailey Alston, Palak Y. Patel, Gaurav Kwatra, Tom Hall, Stephen Thomas, Stephen Taylor, Kirsty Le Doare, Andrew Gorringe

**Affiliations:** 1UK Health Security Agency, Porton Down, Salisbury SP4 0JG, UKandrew.gorringe@ukhsa.gov.uk (A.G.); 2Maternal and Neonatal Vaccine Immunology Research Group, Centre for Neonatal and Paediatric Infection, St George’s, University of London, London SW17 0RE, UK; suzannalim@outlook.com (S.L.); thall@sgul.ac.uk (T.H.);; 3Centers for Disease Control and Prevention, Atlanta, GA 30329, USA; vhd2@cdc.gov (P.M.);; 4Eagle Global Scientific, Atlanta, GA 30341, USA; 5South African Medical Research Council, Vaccines and Infectious Diseases Analytics Research Unit, Faculty of Health Sciences, University of the Witwatersrand, Johannesburg 2050, South Africa; gaurav.kwatra@wits-vida.org; 6Division of Infectious Diseases, Department of Pediatrics, Cincinnati Children’s Hospital Medical Center, University of Cincinnati, Cincinnati, OH 45229, USA

**Keywords:** Group B *Streptococcus*, opsonophagocytosis assay, OPKA, correlates of protection, neonatal antibodies, vaccines, transplacental antibodies

## Abstract

The placental transfer of antibodies that mediate bacterial clearance via phagocytes is likely important for protection against invasive group B *Streptococcus* (GBS) disease. A robust functional assay is essential to determine the immune correlates of protection and assist vaccine development. Using standard reagents, we developed and optimized an opsonophagocytic killing assay (OPKA) where dilutions of test sera were incubated with bacteria, baby rabbit complement (BRC) and differentiated HL60 cells (dHL60) for 30 min. Following overnight incubation, the surviving bacteria were enumerated and the % bacterial survival was calculated relative to serum-negative controls. A reciprocal 50% killing titer was then assigned. The minimal concentrations of anti-capsular polysaccharide (CPS) IgG required for 50% killing were 1.65–3.70 ng/mL (depending on serotype). Inhibition of killing was observed using sera absorbed with homologous CPS but not heterologous CPS, indicating specificity for anti-CPS IgG. The assay performance was examined in an interlaboratory study using residual sera from CPS-conjugate vaccine trials with international partners in the Group B *Streptococcus* Assay STandardisatiON (GASTON) Consortium. Strong correlations of reported titers between laboratories were observed: ST-Ia r = 0.88, ST-Ib r = 0.91, ST-II r = 0.91, ST-III r = 0.90 and ST-V r = 0.94. The OPKA is an easily transferable assay with accessible standard reagents and will be a valuable tool to assess GBS-specific antibodies in natural immunity and vaccine studies.

## 1. Introduction

Group B *Streptococcus* (GBS) is the leading cause of neonatal infection in high income countries and recent estimates suggest that in 2020 almost 400,000 children presented with either early- or late-onset GBS disease, with over 90,000 infant deaths [1]. GBS was also estimated to be a causative factor for 46,200 stillbirths and a risk factor for up to 518,100 pre-term births [1]. Due to this high burden of disease, the World Health Organization (WHO) has proposed priority research and development pathways and preferred product characteristics to accelerate the licensure of GBS vaccines for immunization in pregnancy [2]. GBS vaccines based on capsular polysaccharide-protein conjugates have been in development for many years, including the clinical evaluation of monovalent and multivalent vaccines [3]. More recently, a hexavalent GBS vaccine (comprising serotypes Ia, Ib, II, III, IV and V polysaccharides [4]) has been evaluated in a phase I/II study while phase I results have been reported with an alpha-like protein (Alp) subunit vaccine [5].

Demonstrating efficacy for a GBS vaccine using disease endpoint efficacy trials will be challenging and require significant investment. Due to the low incidence of disease in some regions, trials may require the enrollment of up to 80,000 pregnant women and would likely take several years to complete [6]. An alternative is to identify biomarkers of protection against disease and use an immunological endpoint as evidence for licensure followed by demonstration of effectiveness post introduction [6]. This approach has been used for increased-valency pneumococcal [7] and serogroup B meningococcal disease vaccines [8]. A similar approach for GBS vaccines is likely, as an association between anti-capsular polysaccharide IgG and protection from disease has been shown in animal vaccine studies and human natural immunity studies over many years [2,9].

Key to demonstrating serological correlates of protection are standardized immunoassays that can be reliably used in studies of both natural and vaccine-derived immunity. Anti-GBS capsular polysaccharide IgG has been quantified by many different assays and its function in mediating the opsonophagocytic killing of GBS bacteria has been determined by diverse methods (reviewed by [9]). The lack of standard immunoassays was identified as a key gap following a WHO consultation of GBS vaccine development [10] and led to the establishment of a scientific, industrial and technical consortium (GASTON) to standardize assays in order to define serocorrelates of risk reduction for neonatal and infant GBS disease [9]. This has resulted in the development of a multiplexed (Luminex) immunoassay that measures human serum IgG against six GBS capsular polysaccharides Ia, Ib, II, III, IV and V (Gaylord et al., submitted), which has been evaluated in an interlaboratory study (Le Doare et al., submitted).

As serocorrelates of protection are required before vaccine licensure and introduction, it is important to demonstrate that the functions of antibodies generated by natural exposure or vaccination are the same or closely associated [2]. It is likely that the antibody-mediated opsonophagocytic killing of GBS is the immune mechanism of protection [11], so a parallel activity in the GASTON consortium has been to develop a standardized opsonophagocytic killing assay (OPKA). Two approaches have been used for OPKAs for GBS. Older studies determined the degree of bacterial killing over a fixed incubation period caused by a human serum sample with an endogenous complement [12]. More recent studies assessing pneumococcal vaccines have determined the dilution of antiserum that kills at least 50% of the target bacteria in the presence of an exogenous complement, either with a viable count or fluorescence endpoint [13,14,15].

We have developed an OPKA for GBS that is based on previously described protocols that evaluate the killing mediated by human serum dilutions against representative GBS strains of serotypes Ia, Ib, II, III and V using baby rabbit serum as the exogenous complement source and with viable count determination as the endpoint. We have then evaluated the OPKA in an interlaboratory study and report the results.

## 2. Materials and Methods

### 2.1. Bacteria and Generation of Frozen Bacterial Working Stocks

Group B *Streptococcus* strains representing five serotypes (Ia, Ib, II, III and V) were obtained from the Respiratory and Vaccine Preventable Reference Unit, UK Health Security Agency; Professor Carol Baker, Baylor College of Medicine, Houston; and The National Collection of Type Cultures (NCTC) (Table 1). Strains were cultured on Columbia agar with 5% horse blood (COH; Biomérieux, Lyon, France) and incubated overnight at 37 °C with 5% CO_2_. Colonies were collected and inoculated at an optical density (OD) 600 nm of 0.1 into Todd-Hewitt broth (Oxoid Ltd., Basingstoke, UK). Bacteria were grown for approximately 2 h until OD 600 nm 1.0 (±0.2) was reached. An equal volume of Todd-Hewitt broth containing 50% (*v*/*v*) filter-sterilized glycerol was added and 1 mL aliquots were prepared. Stocks were subsequently stored at −80 °C. Bead stocks of each isolate were also generated using a Microbank cryopreservative system (Pro-Lab Diagnostics, Richmond Hill, ON, Canada) and stored at −80 °C.

### 2.2. Sera

Standard human reference sera (SHRS) for serotypes Ia, Ib, II, III and V were provided by Professor Carol Baker, Baylor College of Medicine, Houston [16]. The SHRS were used during the initial optimization of the assay and as a positive control in all subsequent assays. Serotype-specific rabbit antisera (Ia, #22455; Ib, #22456; II, #22458; III, #22459; V, #22461; Statens Serum Institut (SSI Diagnostica, Hillerød, Denmark) and monoclonal antibodies specific to alpha-like proteins (Rib-N, AlpC-N, Alp1-N and Alp2-N) provided by Per Fischer (MinervaX, Copenhagen, Denmark) were used to assess surface-bound antibodies to GBS isolates [5]. The serum panel used for the interlaboratory study comprised residual sera from clinical trials involving monovalent and multivalent GBS polysaccharide capsule conjugate vaccines kindly provided by Professor Carol Baker (Baylor College of Medicine, Houston, TX, USA), with individual subject consent obtained in each study and approved by the Institutional Review Board for Human Subject Research at Baylor College of Medicine [17,18,19,20,21,22]. A selection of low, medium and high sera was chosen based on previously determined anti-capsular polysaccharide IgG concentrations measured by enzyme-linked immunosorbent assay (ELISA) from immunogenicity studies using monovalent and multivalent CPS-conjugate vaccines [17,18,19,20,21,22,23]. To avoid interference from the endogenous complement in the assay, all sera were heat-inactivated at 56 °C for 30 min prior to testing. All sera were tested blind in the interlaboratory study.

### 2.3. Flow Cytometry

Serotype-specific IgG binding to whole-cell bacteria using GBS typing rabbit antisera (SSI Diagnostica, Hillerød, Denmark) was evaluated by flow cytometry. In a 96-well U-bottom microtiter plate, 2 µL of either homologous or heterologous typing antisera was incubated with 198 µL of 1 × 10^7^ CFU/mL suspension of each isolate at 37 °C for 30 min with shaking at 900 rpm. The plates were centrifuged at 3060× *g* for 5 min and bacterial pellets washed with 2% bovine serum albumin (BSA; Sigma, USA) in a phosphate buffer solution (PBS; Severn Biotech Limited, Kidderminster, UK). Supernatant was removed and the pellets were resuspended in 200 µL of 1:500 fluorescein isothiocyanate (FITC)-conjugated goat anti-rabbit IgG (catalog #111-096-045, Jackson ImmunoResearch, West Grove, PA, USA) at 4 °C for 20 min. The plates were washed twice and the pellets resuspended in 200 µL of filter-sterilized PBS. Acquisition was performed on a CyAn™ ADP flow cytometer (Beckman Coulter, High Wycombe, UK) and analyzed with Summit 4.3 software. Bacteria were gated on forward scatter versus side scatter dot plots. The gating strategy on histogram plots was set based on the negative control samples (bacteria and conjugate, serum-) at a lower limit of 10%. This gating of fluorescence index values >10% indicated populations of positively stained bacteria binding rabbit antisera (Appendix A Appendix A).

### 2.4. HL-60 Cells

HL-60 cells were purchased from American Type Culture Collection (ATCC #CCL-240; ATCC LGC Standards, Teddington, UK) and cultured in an RPMI 1640 medium (Sigma Aldrich, Burlington, MA, USA), 2 mM L-glutamine and 20% heat-inactivated foetal bovine serum (FBS, South American origin; LabTech, Heathfield, UK). Cells were maintained for approximately 6 weeks until a 24 h doubling time was reached. For assays, cells were differentiated into neutrophil-like cells (dHL60) at 5 × 10^5^ cells/mL with 0.8% (*v*/*v*) dimethylformamide (Sigma Aldrich, Burlington, MA, USA) added to fresh culture media as described above and incubated at 37 °C with 5% CO_2_ for 5 days prior to use. To demonstrate successful differentiation, expressions of cell surface markers CD11b, CD55 and CD71 were determined by flow cytometry using fluorescently labelled monoclonal antibodies [24]. Cells expressing ≥55% CD11b, ≥55% CD35 and ≤20% CD71 were then used for OPKA.

### 2.5. Complement

Multiple batches of 3 to 4-week-old baby rabbit complement (BRC; Cedarlane Labs, Burlington, ON, Canada) were blended to make a pool of exogenous complement for assays. Four batches were thawed under running cold water and decanted into appropriate glassware with gentle swirling to preserve complement activity. Aliquots of 1.5 mL were transferred into pre-chilled internally-threaded polypropylene cryovials (Fisher Scientific UK Ltd., Loughborough, UK) and stored at −80 °C until required. All procedures were carried out on ice.

### 2.6. Opsonophagocytic Killing Assay (OPKA)

Stage 1 Opsonization: Serial 2-fold dilutions of heat-inactivated test serum were prepared in a 96-well U-bottom microtiter plate using a filter-sterilized assay buffer (Hank’s Balanced Salt Solution (HBSS) with 0.5 mM MgSO_4_, 0.9 mM CaCl_2_ and 0.1% gelatin; Thermo Fisher Scientific Inc., Burlington, MA, USA). Undiluted serum (40 µL) was pipetted into the first well then 20 µL transferred into 20 µL of assay buffer in the adjacent well, mixed thoroughly with 2-fold dilutions continuing across the plate. For sera with known high concentrations of anti-CPS antibodies, pre-dilutions were prepared on a separate microtiter plate before the 40 µL was transferred to the first well of the assay plate as above. Positive control serum (SHRS) was prepared in the same manner with a starting dilution of 1/128. Each plate also included control wells without serum and without both serum and dHL60 cells. A frozen aliquot of bacteria was thawed and 500 µL was diluted 1:10 in assay buffer and centrifuged at 3060× *g* for 5 min at room temperature. The supernatant was gently removed, taking care not to disturb the bacterial pellet, which was then resuspended in assay buffer. The bacteria were resuspended to a concentration of 2.5 × 10^4^−1.25 × 10^5^ CFU and 20 µL was added to all wells. The plates were incubated for 30 min at 37 °C with shaking at 900 rpm in a microplate incubator shaker.

Stage 2 Phagocytosis: A mixture of dHL60 cells, BRC and assay buffer was prepared as follows: the volume required for 1 × 10^6^ cells/well of dHL60 cells was calculated, then centrifuged at 400× *g* for 5 min, the supernatant was discarded and the cell pellet resuspended in a total volume of 47.5 µL/well (i.e., for 100 wells, 1 × 10^8^ cells were resuspended in 4.75 mL of assay buffer). An amount of 12.5 µL/well of baby rabbit complement was added to the cell suspension and 60 µL of the BRC-dHL60 suspension was pipetted into each well (except for the control wells without serum), bringing the final well volume to 100 µL. A separate suspension of BRC only was prepared (12.5 µL/well of BRC and 47.5 µL/well of assay buffer) and 60 µL was added to the control wells without serum and dHL60 cells. The plates were incubated for 30 min at 37 °C with shaking at 900 rpm, then 10 µL samples from all wells were inoculated onto COH agar plates which were tilted to form streaks and allowed to air dry. The agar plates were incubated overnight at 37 °C with 5% CO_2_. Colonies were enumerated the following day using the Sorcerer colony counter system (Perceptive Instruments, Bury Saint Edmunds, UK).

### 2.7. Titer Determination

For assays to be accepted, the average colony forming units (CFU) in the control wells without serum (which were used to define 100% bacterial survival) must be between 50 to 250 inclusive. The number of bacterial colonies at each dilution was used to calculate the % survival, relative to the control wells without serum. The opsonophagocytic titer was defined as the serum dilution where ≥50% killing was observed. Titers obtained with SHRS were trended for quality control purposes using Levey–Jennings analysis [5]. For each individual test sample, the % survival curves were plotted and titers reported if there were at least two consecutive dilution points with ≥50% killing and at least one dilution point was ≥70% killing. In addition, at least 1 dilution must have >50% survival of the bacteria. Accepted sera were assigned an interpolated opsonophagocytic titer. Test samples not meeting these criteria were repeated at a higher or lower initial dilution as appropriate.

### 2.8. Specificity

Serotype-specific CPSs (kindly provided by GSK plc) were used to absorb the relevant antibodies present in SHRS thereby blocking bacterial opsonization and preventing killing. Briefly, each serotype-specific CPS was diluted 5-fold starting from 100 µg/mL to 0.000256 µg/mL and incubated with SHRS at a concentration known to mediate opsonophagocytic killing for 1 h at 4 °C. Two CPS preparations were used per serotype: the first was homologous to the SHRS used; the second, a heterologous mixture of the remaining 4 serotypes (100 µg/mL per serotype). The opsonophagocytic killing was compared to negative controls comprising SHRS incubated with assay buffer only. Following incubation, 40 µL of each dilution was transferred to a separate microtiter plate and the standard OPKA was carried out.

### 2.9. Interlaboratory Study

Performance and reproducibility of the OPKA was assessed in an interlaboratory study involving several international partners. The four laboratory sites were the UK Health Security Agency (UKHSA), St. George’s University London (SGUL), Centers for Disease Control and Prevention (CDC, USA) and University of the Witwatersrand, Johannesburg, South Africa (Wits). UKHSA provided pooled BRC and serum panels (*n* = 21) to all laboratories, whilst each site obtained bacterial isolates from the accessioned stock at NCTC to generate subsequent in-house glycerol stocks. HL-60 CCL-240 cells were independently obtained from ATCC; routine maintenance and differentiation of cell lines were carried out by each laboratory site. There were some minor differences in equipment and materials used dependent on the laboratory site. These included the use of 5% sheep blood instead of horse blood in Columbia agar plates, different microplate shaker incubators and colony enumeration software. The sera were tested blind at all sites in two independent runs across multiple operators. All data were sent to UKHSA for collation and analyses.

### 2.10. Statistical Analyses

All statistical analyses were carried out using GraphPad Prism version 9.0.0 software. The OPKA optimization assays were carried out in duplicate with the % survival based on serum negative controls, mean values calculated and standard deviation plotted unless stated otherwise. Inhibition assays were performed in triplicate with the % killing based on serum negative controls with mean values and standard deviation. Comparisons of each site in the interlaboratory study were analyzed by Pearson’s correlation method.

## 3. Results

### 3.1. Strain Selection

Forty-nine GBS isolates representing serotypes Ia, Ib, II, III and V (*n* = 11, 10, 9, 10 and 9, respectively) were examined for binding of serotype-specific IgG using GBS typing rabbit antisera (SSI Diagnostica, Hillerød, Denmark) by flow cytometry. Each strain was evaluated against both homologous and heterologous typing antisera. All isolates showed strong binding of antisera specific for their designated serotype with minimal cross-reactivity to heterologous serotypes (Figure 1).

Several isolates were observed to have atypical characteristics, either where no binding of anti-Alp protein monoclonal antibodies occurred or where increased sensitivity to the complement and dHL60s was observed in the absence of specific antisera. These strains were excluded from further testing (Figure 1). In addition, strains were evaluated for their performance in a prototype OPKA with the serotype matched SHRS (Figure 2). The survival curves of some isolates were outliers when compared to the full panel of strains examined, indicating their unsuitability as representative strains. The final isolate panel selected for inclusion in the standard assay was based on representative performance in the OPKA (bold lines on Figure 2), binding of the expected typing antiserum and no sensitivity to the complement and dHL60 cells in the absence of specific antiserum. These strains are further described in Table 1 and have been deposited with the National Collection of Type Cultures (NCTC, Salisbury, UK), making them available to purchase.

### 3.2. Assay Development and Optimization

Multiple assay parameters were examined to determine their effects on the final opsonophagocytic titer. These included the assay temperature (25 °C or 37 °C), length of incubation during the phagocytic stage (15, 30, 45 and 60 min), concentration of the baby rabbit complement used (3, 6, 12.5 and 25%) and use of individual or pooled batches of the baby rabbit complement (Figure 3). Of the various parameters, only a lower baby rabbit complement concentration affected titers by greater than a single dilution (Figure 3C). Use of a pooled batch of complement eliminated the variation observed with individual batches (Figure 3D). The finalized protocol is represented graphically in Figure 4 with an initial opsonic stage of antibody binding to bacteria during a 30 min incubation at 37 °C, followed by a phagocytic stage with dHL60 cells and 12.5% pooled baby rabbit complement for 30 min at 37 °C (Figure 4).

### 3.3. Assay Specificity

Inhibition experiments using purified capsular polysaccharide were performed to determine assay specificity. For each serotype, a 1 in 5 dilution series (starting at 100 µg/mL) of either the homologous CPS or a mixture of the other four CPSs (at 100 µg/mL per serotype) was prepared and then incubated with the corresponding SHRS added at a known titer for opsonophagocytic killing. Negative controls included incubation with phosphate-buffered solution (PBS) instead of serum. GBS of the matched serotype was added and the remainder of the assay was performed as above. For all serotypes, we observed inhibition of killing only when the homologous CPS was used (Figure 5). The highest level of inhibition was observed in ST-Ia followed by ST-II, ST-Ib, ST-III and ST-V. The heterologous CPS mix did not inhibit antibody-mediated opsonophagocytic killing (shown by the bold black line) at any dilution. Of note, complete inhibition (100%) of killing was not achieved by the addition of serotype-matched CPS, indicating the presence of other antibody specificities in the SHRS that may also mediate killing.

### 3.4. Assay Sensitivity

The sensitivity of the assay was determined using previously determined SHRS anti-CPS IgG concentrations described by [16]. The SHRS for each serotype was used in the assay at a starting dilution of 1/128 which corresponded to the starting concentrations of anti-CPS IgG as denoted on the X-axis for each serotype. The lowest serum dilution where 50% killing was detectable corresponded with anti-CPS IgG concentrations of 0.00356 µg/mL, ST-Ia; 0.00365 µg/mL, ST-Ib; 0.00173 µg/mL, ST-II; 0.00370 µg/mL, ST-III and 0.00165 µg/mL, ST-V (Figure 6).

### 3.5. Interlaboratory Study

Four sites across three continents participated in an interlaboratory study to assess the performance of the optimized opsonophagocytic killing assay. The sera were tested twice in two independent assay runs with operators blinded to the anti-CPS IgG concentration of serum samples. All sites were provided by UKHSA with the same baby rabbit complement pool and serum panel. Each laboratory independently cultured bacteria and created their own working stocks in-house; routine maintenance of HL60 cells and differentiation were also carried out at each location. Minor differences between laboratories included the use of sheep-blood agar plates (instead of horse-blood), general laboratory equipment and colony enumeration. Data for all serotypes were collected for laboratories 1–3 whilst only serotype III data were available for laboratory 4. All data were collated and analyses were performed at UKHSA. For all serotypes, we observed similar distributions of OPKA titers obtained at all laboratories with geometric means shown in Figure 7A. Laboratory 2 was found to consistently give significantly lower titers across all sera and serotypes tested than laboratories 1 and 3. Despite this, pairwise comparisons between laboratories using Pearson’s analysis showed strong correlation for all serotypes, ranging from r = 0.86 to 0.97 (Figure 7B, Table 2). An average correlation calculated for each serotype was determined as: ST-Ia, r = 0.88; ST-Ib, r = 0.91; ST-II, r = 0.91; ST-III, r = 0.90 and ST-V, r = 0.94.

A secondary analysis was also performed to investigate whether testing of sera in duplicate produced an interpolated titer that was within +/− one dilution. All titers determined across all four laboratory sites for each serum were observed to be within a single dilution with the exception of one result from laboratory 3. From this observation, we recommend that the duplicate testing of sera is not necessary using these assay conditions.

## 4. Discussion

We report the development of a standardized opsonophagocytic killing assay (OPKA) for GBS and the first interlaboratory study with a GBS OPKA. The use of standardized assays will permit comparison of assay results across the GBS vaccine research community and will allow bridging to existing assays and to assays that may be developed using high throughput approaches, including those that do not rely on a viable count endpoint. We have shown that highly similar results can be obtained, even with phagocytic cells and bacteria cultured in different laboratories. This will be important in establishing accepted antibody thresholds associated with risk reduction of invasive disease and will link studies of naturally acquired and vaccine-induced immunity. The OPKA has the advantage of using live GBS bacteria as the target so the assay can be used to assess the function of antibodies generated against both CPS and protein antigen targets. To that end, we assessed the expression of alpha-like proteins to ensure the chosen strains were suitable to assess a vaccine in clinical development [5], in addition to CPS-conjugate vaccines.

The importance of the assessment of functional antibodies was emphasized by the considerations from the World Health Organization consultation on the role of immune correlates of protection on the pathway to licensure of GBS vaccines [2]. The authors noted the intention to extrapolate results from immunity studies in conditions of natural exposure to immunity resulting from vaccination and that this assumes that the protective mechanisms are the same. Assessing both antibody quantity and function provides the link between these two settings until any differences between naturally or vaccine-induced immunity are better understood. Furthermore, it is understood that IgM contributes to OPKA killing with maternal sera but only IgG is transferred to the infant. Thus, it will be important to include infant sera in studies to determine protective thresholds and not only analyses of sera from mothers. Antibody function can also be used to ensure new vaccines based on novel antigens or different conjugation technologies provide antibodies with equivalent functions.

Key to the development of the assay was the selection of suitable target strains that were representative of the behavior of current disease-causing isolates in the assay. We selected strains that produced survival curves with SHRS that were like most strains tested. Strains that were more sensitive or resistant to killing were excluded. The selected strains have been deposited in a culture collection and are available to any group wishing to perform the assay. We have demonstrated that the assay is robust with little effect of changes in the incubation times or temperature. The concentration of the complement was important for antibody function and some variation in assay performance between batches was observed. However, the pooling of batches overcame this variation and produced consistent results.

It is important to note that the assay is highly sensitive with 50% killing observed at very low concentrations of anti-CPS IgG (1.65–3.70 ng/mL depending on serotype). This sensitivity will allow quantification of OPKA activity in samples with low antibody concentrations, such as those from natural immunity studies, which may include dried blood spot samples [25] that are diluted 10-fold during elution from the matrix before analysis.

Various opsonophagocytosis assays have been used to assess the anti-GBS antibody function. Recently, a GBS multiplexed opsonophagocytic assay (OPA or MOPA) has been described [13], which was based on the pneumococcal MOPA [26,27]. This approach was not adopted in this study. Instead, we developed an easily transferable standard assay that did not require special equipment and could use commercially available agar plates. Our approach ensures that each laboratory can use locally available growth media and equipment while the assay fundamentally remains the same. We also wanted to ensure that the strains which were selected were representative of recent disease-causing isolates. The MOPA requires antibiotic-resistant strains to be developed by multiple passages on media containing increasing concentrations of antibiotics. The Choi study [26] does not describe why the selected strains were chosen, nor if the titers obtained were unchanged by the acquisition of resistance. Other approaches have used fluorescently labelled bacteria with uptake by phagocytic cells determined by flow cytometry [28]. This approach was refined [14] where fixed bacteria were labelled with pHrodo, a fluorescent dye that increases in fluorescence in the acidic conditions present in the phagocytic endosomal compartment. Both approaches require flow cytometry facilities and do not evaluate antibody-mediated killing of the target bacteria.

A limitation of the current study is that it was performed during the COVID-19 pandemic and was greatly delayed by pandemic-related laboratory activities. As a result of this, one laboratory was only able to complete analysis of one serotype. Despite this, excellent agreement was observed between laboratories in the interlaboratory study.

The next steps for this approach are to assign OPKA titers to the SHRS serotype-specific pools. In addition, efforts are underway to develop a large volume international human serum reference standard with assigned serotype-specific IgG concentrations and OPKA titers could also be assigned. A large volume serum panel would also be of immense value to ensure assay performance in different laboratories remains consistent. We will also add additional strains to the panel to represent vaccines in development e.g., serotype IV [4,29,30]. Using the standardized method reported here, evaluation of the less common serotypes of GBS, which are more prevalent in Asia [31,32,33], will be pivotal should serotype replacement occur in the future. Enzyme-linked immunosorbent assays (ELISA) have been used extensively to study the immune correlates of protection for pneumococcal CPS vaccines, especially in pediatric populations [34,35]. However, older adults tend to have high antibody concentrations prior to vaccination [36] and ELISA has failed to predict clinical protection for some serotypes [37]. OPA mimics the in vivo mechanism of protection against gram-positive capsulated bacteria [11] and has been used to assess pneumococcal vaccines using single strain assays [38] or a multiplexed assay (MOPA) [39,40] with good agreement between laboratories despite minor variations in the methods.

There has been good recent progress in the clinical evaluation of both CPS-conjugate [4] and protein antigen vaccines [5] and standard immunoassays to quantify anti-CPS IgG and now OPKA. These assays are now available to help define protective antibody thresholds and thus facilitate vaccine licensure and introduction.

## 5. Conclusions

We have developed a simple and easily transferable OPKA that uses standard strains and reagents. This will be a valuable tool for the evaluation of anti-GBS immunity across natural immunity and vaccine studies.

## Figures and Tables

**Figure 1 vaccines-11-01703-f001:**
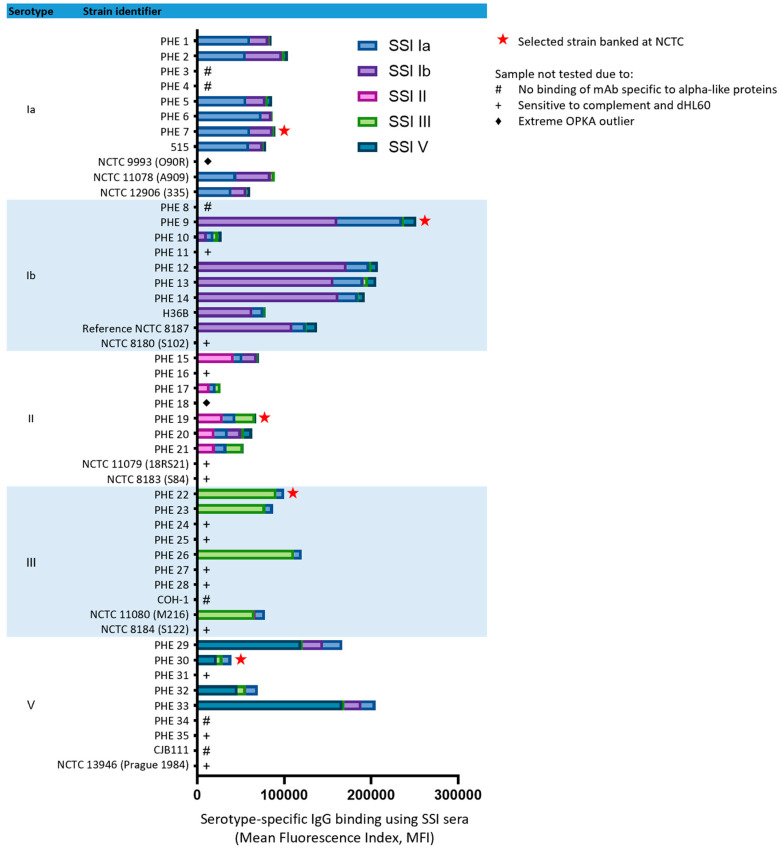
Binding of serotype-specific rabbit antisera (SSI Diagnostica, Hillerød, Denmark) to 49 GBS isolates representing serotypes Ia, Ib, II, III and V was determined by flow cytometry. Each strain was evaluated against both homologous and heterologous typing antisera in duplicate (mean fluorescence values and standard deviations are listed in Appendix A).

**Figure 2 vaccines-11-01703-f002:**
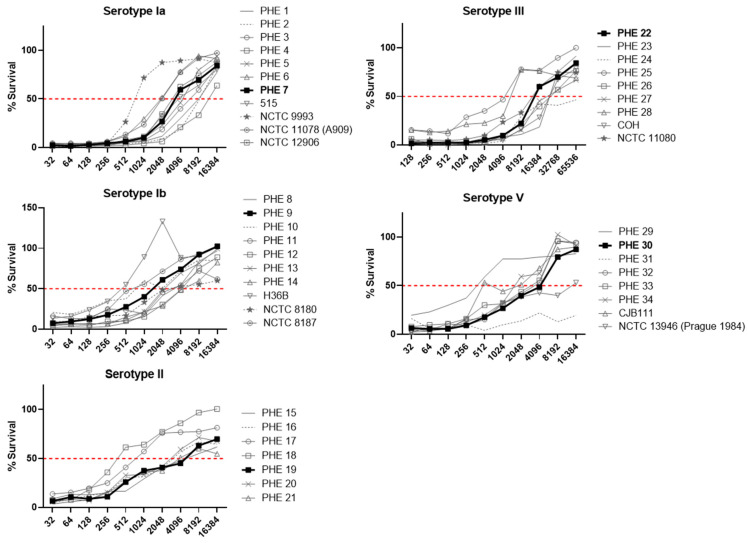
Performance of 49 GBS isolates in OPKA using serotype-matched standard human reference sera (SHRS). The selected isolate is shown in bold. Experiments were carried out in duplicate with plotted mean values.

**Figure 3 vaccines-11-01703-f003:**
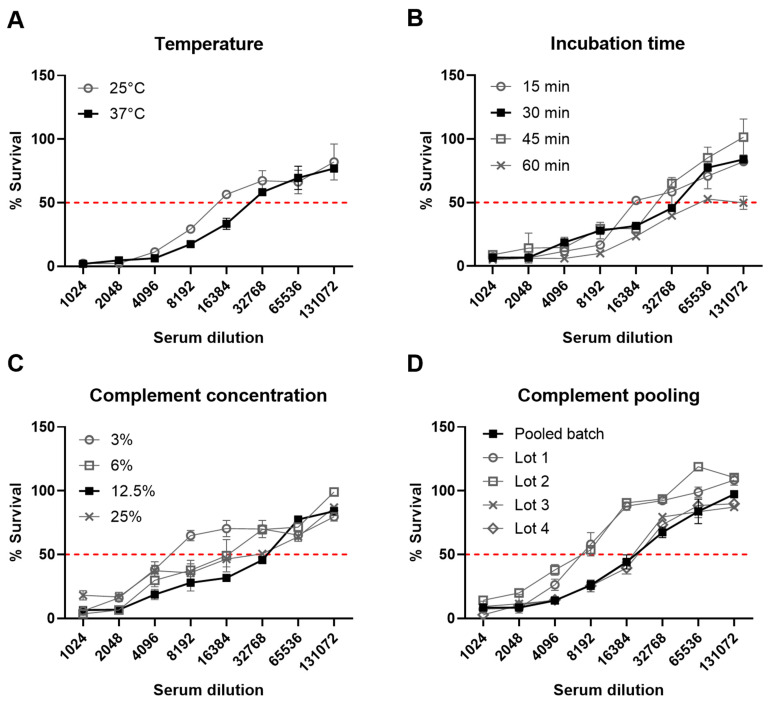
During assay optimization, several parameters were investigated including (**A**) incubation temperature, (**B**) incubation time, (**C**) complement concentration and (**D**) use of individual or pooled batches of complement. Experiments were carried out in duplicate with geometric mean and standard deviation shown.

**Figure 4 vaccines-11-01703-f004:**
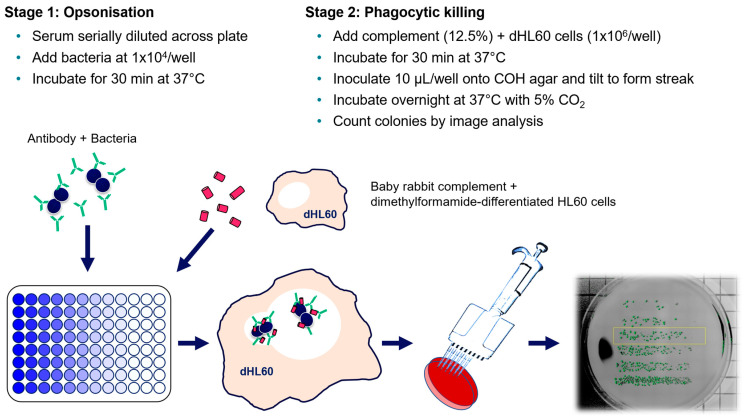
Schematic of optimized opsonophagocytic killing assay (OPKA). The assay comprises two stages: test serum (serially diluted) is incubated with bacteria to allow for opsonization to occur; followed by addition of baby rabbit complement and dHL60 cells to initiate phagocytic killing. COH agar plates are inoculated in the final step by plating 10 µL of assay mixture and tilting to form streaks, followed by an overnight incubation. Colonies (highlighted green by image analysis software) are enumerated using manual or digital image analysis methods. Killing titers are calculated as the 50% reciprocal dilution point compared to the average CFU in serum-free controls.

**Figure 5 vaccines-11-01703-f005:**
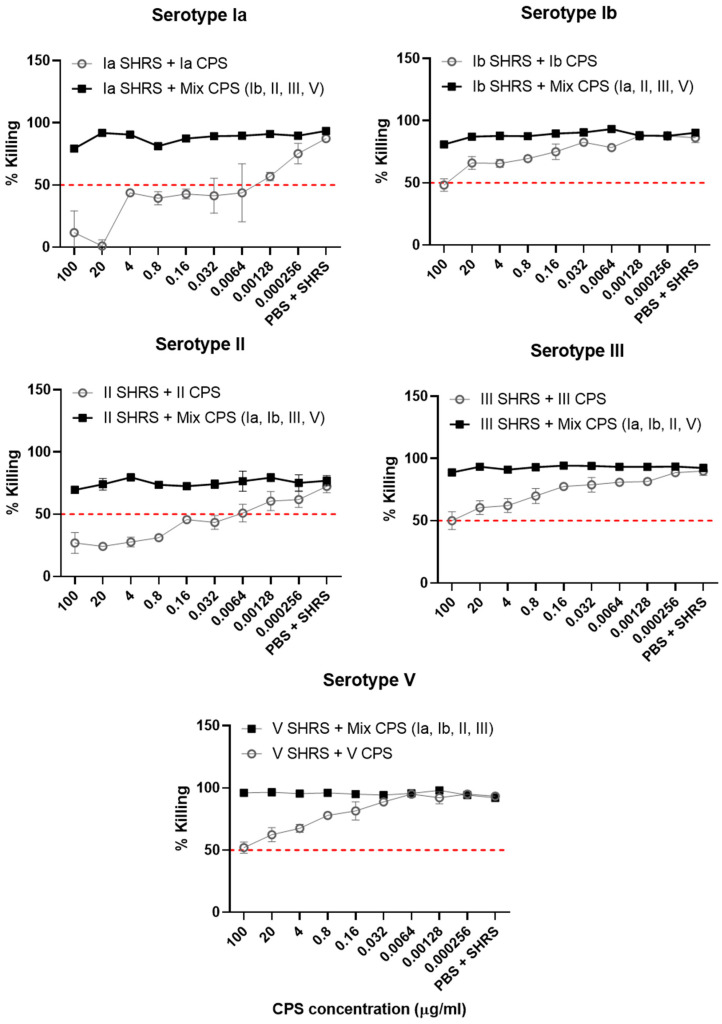
Homologous and heterologous CPS mixes were prepared (1 in 5 dilution series) and pre-incubated with SHRS at a known killing titer. The corresponding GBS serotype isolate was added and the remainder of the assay was carried out. Inhibition experiments were carried out in triplicate with mean and standard deviation shown.

**Figure 6 vaccines-11-01703-f006:**
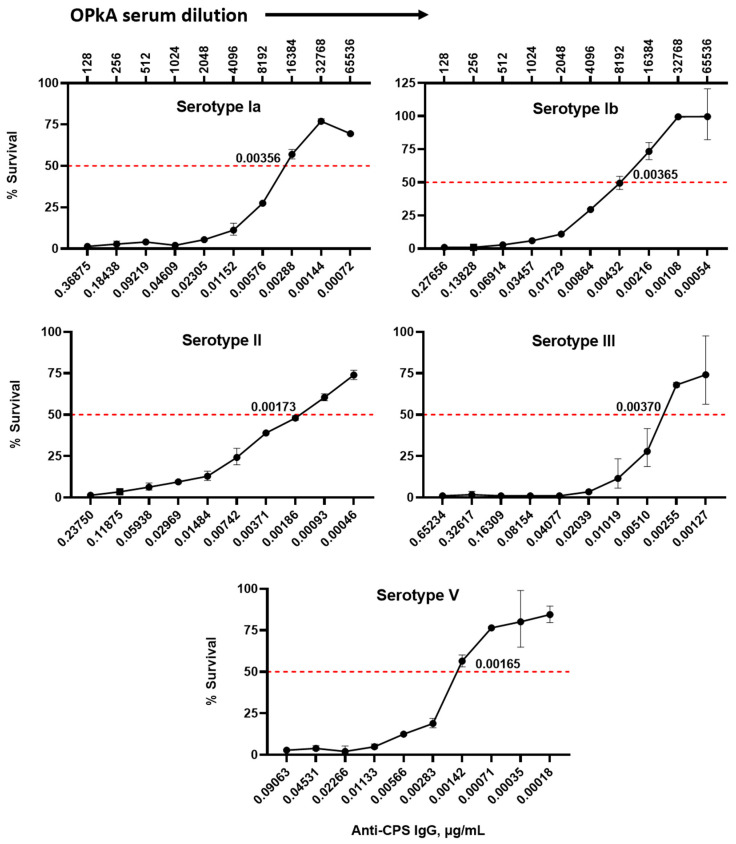
Determination of anti-CPS IgG concentration required to mediate 50% killing. Anti-capsular polysaccharide antibody concentrations denoted on the x-axis were calculated corresponding to the starting dilution of SHRS used for the OPKA. Experiments were carried out in duplicate with geometric mean and geometric standard deviation shown.

**Figure 7 vaccines-11-01703-f007:**
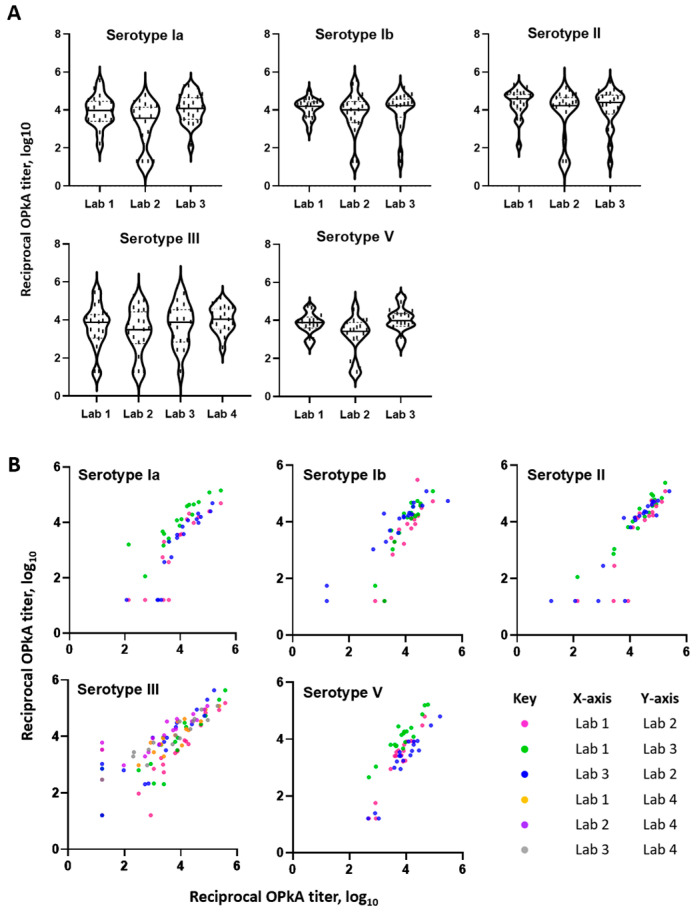
Performance of OPKA assessed by interlaboratory study. Four sites across three continents participated and tested the same proficiency serum panel with multiple operators. (**A**) Distribution of OPKA titers obtained are shown for each laboratory and serotype. Samples were tested in duplicate with geometric mean values plotted. The solid black line represents the population median and the dashed black line represent 25% and 75% percentiles. (**B**) Pairwise comparisons between laboratories by Pearson’s analysis demonstrated strong correlations, as detailed in Table 2.

**Table 1 vaccines-11-01703-t001:** Clinical GBS isolates collected from neonatal disease cases were selected to represent 5 serotypes and accessioned at the National Collection of Type Cultures (NCTC).

NCTC Identification	Serotype	Clonal Complex	Site and Date of Isolate	Surface Protein Expression
14,094	Ia	23	Blood culture, 2014	Alp1-N
14,092	Ib	8	Blood culture, 2014	AlpC-N
14,093	II	28	Blood culture, 2014	Rib-N
14,091	III	17	Blood culture, 2009	Rib-N
14,095	V	1	Blood culture, 2014	Alp2-N

**Table 2 vaccines-11-01703-t002:** Pairwise comparisons of opsonophagocytic killing titers between each laboratory was carried out and Pearson’s analysis applied.

Laboratory Comparisons	Pearson Correlation (r)				
X, Y	ST Ia	ST Ib	ST II	ST III	ST V
Lab 1, Lab 2	0.86	0.89	0.87	0.95	0.95
Lab 1, Lab 3	0.89	0.90	0.97	0.88	0.93
Lab 3, Lab 2	0.90	0.95	0.88	0.86	0.94
Lab 1, Lab 4	-	-	-	0.88	-
Lab 2, Lab 4	-	-	-	0.87	-
Lab 3, Lab 4	-	-	-	0.95	-

## Data Availability

Data from this study are available on request to the GASTON consortium, https://cnpi-vaccinology.com/gaston/.

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
