# Peer review of "Development of A Standardized Opsonophagocytosis Killing Assay for Group B Streptococcus and Assessment in an Interlaboratory Study"

_vaccines, 2023, doi:10.3390/vaccines11111703_

Round 1

Reviewer 1 Report

Comments and Suggestions for Authors

Line 112-113: “A selection 112 of low, medium and high sera was chosen based on their previously determined anti-113 capsular polysaccharide IgG concentrations” Please mention the assay how was the IgG concentration determined.

Line 116: 2.3. Flow cytometry: Please add a reference or include some dot plots showing the gating strategy for the flow cytometric analysis. Did you include isotype control staining?

What is the rational behind using BRC rather than stabilized Guinea pig complement?

Please describe the statistical analyses in detail

Did you inactivate (heating to 56 C) the tested sera to avoid the interference through endogenous complement ? May be it is important to evaluate the impact of endogenous complement concentration on the assay performance?

For all figure legends, please include statistical information (number of experiments, replicates, and statistical test used)

Author Response

Dear Reviewer,

Thank you very much for taking the time to review our manuscript for “Development and interlaboratory study of a standardised opsonophagocytosis killing assay for group B Streptococcus”. Please find the detailed responses below and the corresponding revisions/corrections highlighted in the re-submitted files.

  • Line 112-113: “A selection of low, medium and high sera was chosen based on their previously determined anti-capsular polysaccharide IgG concentrations” Please mention the assay how was the IgG concentration determined. References 17-22 have been added to describe the enzyme-linked immunosorbent assay (ELISA) previously used to determine IgG concentrations.
  • Line 116: 2.3. Flow cytometry: Please add a reference or include some dot plots showing the gating strategy for the flow cytometric analysis. Did you include isotype control staining? Dot plots have been added to the supplementary materials and control staining mentioned in main text (lines 136-140).
  • What is the rationale behind using BRC rather than stabilized Guinea pig complement? Baby rabbit complement is widely used for functional killing assays in the Streptococcus field and was selected to build upon experiences of previously published studies (references 13, 14, 24, 26 and 32).
  • Please describe the statistical analyses in detail. More detail has been added to section 2.10 (lines 242-246).
  • Did you inactivate (heating to 56°C) the tested sera to avoid the interference through endogenous complement? Maybe it is important to evaluate the impact of endogenous complement concentration on the assay performance? Yes, sera were heat-inactivated, therefore endogenous complement should have no impact on assay performance – added to manuscript section 2.2 (lines 121-122).
  • For all figure legends, please include statistical information (number of experiments, replicates, and statistical test used). Information has been added to manuscript, see legends for figures 1 to 7.

We hope that you will find the addition of supplementary materials and revisions to the manuscript suitable and thank you for the helpful feedback.

Reviewer 2 Report

Comments and Suggestions for Authors

Line 31:  "titers" rather than titres.  Line 47 : Should be "serotypes". Line 48 : Polysaccharides "have" been ; also "phase 1". Line 98: bacterial "aliquots.   Line 113: high sera "were" chosen. Line 195 : repeated at started a higher ??

Figure 1 is very clear. Figure 2 is also clear as to the isolates that were selected; (similarly with Figure 3). Figure 4 has a problem "the last part is not clear "please clarify". I have not detected any problems with Figures 5 or 6.

Figure 7 "A" and "B" are well presented.

Page 13 : Lines 422-435, there appears to be a Table which is incomprehensible ? Please clarify or eliminate.

Other than these minor problems, this is an excellent collaborative effort to standardize a killing assay for group B Streptococcus across various laboratories in different countries. 

This is an excellent collaborative effort to improve killing of group B Streptococcus.

Author Response

Dear Reviewer,

Thank you very much for taking the time to review our manuscript for “Development and interlaboratory study of a standardised opsonophagocytosis killing assay for group B Streptococcus”. Please find the detailed responses below and the corresponding revisions/corrections highlighted in the re-submitted files.

  • Line 31: "titers" rather than titres. Our draft was written in British English and we have amended spelling to reflect American English.
  • Line 47: Should be "serotypes". Revisions have been made to manuscript (line 50).
  • Line 48: Polysaccharides "have" been – sentence subject is “vaccine” so we have used the singular “has”. Also "phase 1". Phase 1 amended to “Phase I” for consistency (line 51).
  • Line 98: bacterial "aliquots”. Wording has been revised in manuscript (line 101).
  • Line 113: high sera "were" chosen. Amended in manuscript (line 117).
  • Line 195: repeated at started a higher?? Wording has been revised in manuscript (line 213).
  • Figure 1 is very clear. Figure 2 is also clear as to the isolates that were selected; (similarly with Figure 3). Figure 4 has a problem "the last part is not clear "please clarify". I have not detected any problems with Figures 5 or 6. Thank you for these comments – we have adjusted figure 4 by the addition of an extra image to illustrate the plating technique and added more descriptive text to the figure legend.
  • Figure 7 "A" and "B" are well presented. Thank you for these comments.
  • Page 13: Lines 422-435, there appears to be a Table which is incomprehensible? Please clarify or eliminate. The table has been deleted.
  • Other than these minor problems, this is an excellent collaborative effort to standardize a killing assay for group B Streptococcus across various laboratories in different countries. This is an excellent collaborative effort to improve killing of group B Streptococcus. We are grateful for these encouraging comments.

We hope that you will find the revisions to the manuscript suitable and thank you for the helpful feedback.

Reviewer 3 Report

Comments and Suggestions for Authors

The paper describes a method to standardize opsonophagocytosis killing assay for group B Streptococcus (GBS). This assay is intended to determine the protective antibody titer following GBS vaccination. The assay is built on the killing of various serotypes of GBS using serotype-specific rabbit antisera, baby rabbit complement (BRC) and differentiated HL60 cells. The antibody titer was dependent on low BRC concentrations. Already at low concentration of anti-capsular polysaccharide (CPS) IgG, a 50% killing was observed. English editing is required. (Several grammatically problematic sentences).

From the Abstract it is difficult to follow how they performed the assay and how they determined the antibody titer. The abstract does not describe which blood samples were tested in the interlaboratory study, and which standard test the results were compared to.  The abstract should be more concise.

Line 52: correct to "enrollment".

Line 107: Add the catalog number of sera and all reagents.

Line 108: All abbreviations should be spelled in full name first time used. The authors need to state what is the epitopes Rib-N, AlpC-N, Alp1-N and Alp2-N.

Line 110: State the Helsinki Approval for using the residual sera from clinical trials. And the approval number of the clinical trials.

Line 120: The exact amount of bacteria and volume should be stated.

Line 123: The concentration of the secondary antibody should be stated. And the catalog number of the antibody.

The flow cytometry should be tested also on human serum samples to determine the antibody titer.

Line 129 – the ATCC number should be stated.

Line 133: Was DMF added to full medium or serum-free medium? Please state.

Line 144: correct to: "Serial 2-fold dilutions"

Line 147: What do you mean with "neat sera"?

From line 155: The bacteria were used directly from the frozen stock into the assay. It means – no recovery after thawing. If the bacteria were allowed to recover before the assay, would similar results be obtained?

Line 156: Please state the OD:CFU conversion.

Line 169: Please demonstrate how the 10 microliters were spread on the plate.

Line 222: The statistical assay should be mentioned and how this was performed.

In Figure 4: Why are the colonies green?

Lines 310-315 – Should be shortened – as this is a repetition of method section.

For analysis of GBS IgG titers in human serum – is the rabbit complement similar active as with rabbit IgG? Is the rabbit complement necessary for human GBS IgG?

Comments on the Quality of English Language

English editing required. 

Author Response

Dear Reviewer,

Thank you very much for taking the time to review our manuscript for “Development and interlaboratory study of a standardised opsonophagocytosis killing assay for group B Streptococcus”. Please find the detailed responses below and the corresponding revisions/corrections highlighted in the re-submitted files.

  • The paper describes a method to standardize opsonophagocytosis killing assay for group B Streptococcus (GBS). This assay is intended to determine the protective antibody titer following GBS vaccination. The assay is built on the killing of various serotypes of GBS using serotype-specific rabbit antisera, baby rabbit complement (BRC) and differentiated HL60 cells. The antibody titer was dependent on low BRC concentrations. Already at low concentration of anti-capsular polysaccharide (CPS) IgG, a 50% killing was observed. English editing is required. (Several grammatically problematic sentences). We have carried out English editing, all authors are native English speakers.
  • From the abstract it is difficult to follow how they performed the assay and how they determined the antibody titer. The abstract does not describe which blood samples were tested in the interlaboratory study, and which standard test the results were compared to. The abstract should be more concise. The abstract has been rewritten to include more information regarding the assay we have developed, titer assignment and the use of residual clinical trial sera for the interlaboratory study (lines 20-33).
  • Line 52: correct to "enrollment". Our draft was written in British English and we have amended spelling to reflect American English.
  • Line 107: Add the catalog number of sera and all reagents. Added to manuscript (lines 108-109).
  • Line 108: All abbreviations should be spelled in full name first time used. The authors need to state what is the epitopes Rib-N, AlpC-N, Alp1-N and Alp2-N. We have added a reference for the relevant paper which describe these epitopes (reference 5).
  • Line 110: State the Helsinki Approval for using the residual sera from clinical trials. And the approval number of the clinical trials. Consent and approval details have been added (line 114-116).
  • Line 120: The exact amount of bacteria and volume should be stated. Amended in manuscript (line 126).
  • Line 123: The concentration of the secondary antibody should be stated. And the catalog number of the antibody. Amended in manuscript (line 130-131).
  • The flow cytometry should be tested also on human serum samples to determine the antibody titer. Serotype-specific rabbit antisera was used for the purpose of confirming the serotype of each isolate. Human serum samples contain antibodies with multiple specificities and would not have been appropriate for this test.
  • Line 129 – the ATCC number should be stated. The ATCC number has been added (line 141).
  • Line 133: Was DMF added to full medium or serum-free medium? Please state. Amended in manuscript (line 147)
  • Line 144: correct to: "Serial 2-fold dilutions". Amended in manuscript (line 157).
  • Line 147: What do you mean with "neat sera"? Sera is undiluted, amended in manuscript (line 160).
  • From line 155: The bacteria were used directly from the frozen stock into the assay. It means – no recovery after thawing. If the bacteria were allowed to recover before the assay, would similar results be obtained? More details have been added to describe preparation of bacteria for use in the assay (line 169-171).
  • Line 156: Please state the OD:CFU conversion. We do not carry out an OD:CFU conversion. We determine the bacterial concentration using a sighting assay and have added a description of this (lines 171-175).
  • Line 169: Please demonstrate how the 10 microliters were spread on the plate. We have added extra images to figure 4 for clarification and more descriptive text to the legend.
  • Line 222: The statistical assay should be mentioned and how this was performed. More detail has been added to section 2.10 (lines 240-245).
  • In Figure 4: Why are the colonies green? The image is a screenshot of the software program used to count colonies and the program highlights the colonies as green. This is clarified in the legend of figure 4.
  • Lines 310-315 – Should be shortened – as this is a repetition of method section. We understand that there is slight repetition, however, it is important to highlight the procedure and minor differences that occurred at each testing site to demonstrate the robustness of our assay.
  • For analysis of GBS IgG titers in human serum – is the rabbit complement similar active as with rabbit IgG? Is the rabbit complement necessary for human GBS IgG? We have observed that rabbit complement is similarly as active using rabbit IgG, however, this was not within the scope of this paper. Our assay measures antibody titers derived from human serum whilst using baby rabbit complement as an exogenous complement source. Baby rabbit complement is widely used for functional killing assays in the Streptococcus field and was selected to build upon experiences of previously published studies (references 13, 14, 24, 26 and 32).

We hope that you will find the revisions to the manuscript suitable and thank you for the helpful feedback.

Round 2

Reviewer 3 Report

Comments and Suggestions for Authors

The title is not concise and does not tell the purpose of the assay.

Maybe: Development of a standardized opsonophagocytic killing assay for detecting serotype-specific antibodies to group B Streptococcus: An interlaboratory study

The paper needs extensive English scientific editing. There are so many inaccuracies – and the person that wrote the manuscript seems not to have understood the assay thoroughly or is not familiar with the scientific concepts.

The abstract needs to be presented in a much clearer and more precise manner.

Line 17: correct to "bacterial clearance"

Line 18: "protecting against"

Line 19 is clumsy – you can remove "support studies" and directly write: "to determine" and "is helpful in vaccine development"

Line 25: I think you should have a point instead of a comma.  Followed by sentence: " % bacterial survival was calculated" 

The question is if the calculation is compared to serum negative controls or the input? Since in the next sentence it says that some isolates are killed by BRC-dHL60 even in the absence of antibody which theoretically be set to 100% according to sentence in line 25. Please clarify in the text this issue.

Line 26: I think you meant "killed by" instead of "sensitive to" .

Line 29: You have to state if it is low or high sensitivity.

Line 28: maybe better to write the numbers in ng/mL?

Line 30: Instead of "demonstrating" I would write "indicating". Again specify what kinds of "specificity".

There are also some grammar issues in Introduction.

Line 125: Please state if you used flat-bottomed or round-bottomed microplate.

Line 147: Did you mean "as described above"?

Line 151: Instead of "create" (meaning make something from nothing) I would suggest to write "make".

Line 162: "two-fold dilutions" (instead of "doubling dilutions").

Line 172: What do you mean with "sighting assays"? Please describe better in the text.

Line 175: add "were" before "added".

Line 177: Please define the composition of OPKA buffer. (In line 184 – the authors used the concept "assay buffer"). Be consequent.

Line 186: Please describe the composition of the controls.  It is not clearly presented. The minus sign is mistakenly taken as a hyphen. Please write in full name (minus or without).

Line 187: Define COH.

The method section should be written in a more scientific way. Very clumsy and cumbersome.

The calculations should be written in equations.

How is the survival curve plotted. Please write the formula.

Line 213: What do you mean with " U/N"?

Line 216: Who is GSK?

Line 218: Do you mean 5-fold dilution?

Figure 1: In the figure: "No protein mAb binding" – Do you mean "no mAb binding?" So how were these serotypes defined if the serotype-specific antibodies didn't bind to the bacteria?

Line 216:  It means that the assay only works with specific strains? Meaning a limitation of the assay?

What could be the reason for the phenomenon of outliers?

Line 273: " opsonophagocytic"

Are the representative bacteria available for the general audience?

Line 288: please reformulate – you can not say "facilitated by" – the dHL60 is the phagocytic cells, not the facilitator.

In Figure 4: correct to "dimethylformamide-differentiated HL60 cells"

How do you know if the HL60 cells are completely differentiated?

Comments on the Quality of English Language

There are several grammatical and incomplete sentences.

Also the scientific presentation has to be improved.

Author Response

Dear Reviewer,

Thank you very much for taking the time to review our manuscript for “Development and interlaboratory study of a standardised opsonophagocytosis killing assay for group B Streptococcus”. Please find the detailed responses below and the corresponding revisions/corrections highlighted in the re-submitted files.

  • The title is not concise and does not tell the purpose of the assay. Maybe: Development of a standardized opsonophagocytic killing assay for detecting serotype-specific antibodies to group B Streptococcus: An interlaboratory study. We respectfully disagree with the reviewer’s suggested change to the title. The objective of the consortium was to develop an assay which can also cover future vaccine formulations, including those currently in clinical trials that are based on alpha-like fusion proteins. Thus, it is not solely for determination of the function of anti-capsular polysaccharide-specific antibodies. We state in our abstract that it is a tool to assess GBS-specific antibodies which can include natural immunity and vaccine sera.
  • The paper needs extensive English scientific editing. There are so many inaccuracies – and the person that wrote the manuscript seems not to have understood the assay thoroughly or is not familiar with the scientific concepts. We are grateful for the reviewer’s thorough review of the manuscript which has allowed us to reconsider the text of the method section. This has been extensively revised and made clearer to follow.
  • The abstract needs to be presented in a much clearer and more precise manner. We have carried out some minor revisions to the abstract in response to this comment.
  • Line 17: correct to "bacterial clearance". Wording has been revised as suggested (line 18).
  • Line 18: "protecting against". We do not believe that this minor change is required.
  • Line 19 is clumsy – you can remove "support studies" and directly write: "to determine" and "is helpful in vaccine development". The sentence has been revised as suggested (line 20).
  • Line 25: I think you should have a point instead of a comma.  Followed by sentence: " % bacterial survival was calculated". We have revised the sentence as suggested (line 26).
  • The question is if the calculation is compared to serum negative controls or the input? Since in the next sentence it says that some isolates are killed by BRC-dHL60 even in the absence of antibody which theoretically be set to 100% according to sentence in line 25. Please clarify in the text this issue. We have deleted this sentence as it is no longer required in the revised abstract.
  • Line 26: I think you meant "killed by" instead of "sensitive to". We have deleted this sentence as it is no longer required in the revised abstract.
  • Line 29: You have to state if it is low or high sensitivity. We have deleted this sentence as it is no longer required in the revised abstract.
  • Line 28: maybe better to write the numbers in ng/mL? Wording has been revised as suggested (line 26).
  • Line 30: Instead of "demonstrating" I would write "indicating". Again specify what kinds of "specificity". Wording has been revised as suggested (lines 27-28).
  • There are also some grammar issues in Introduction.
  • Line 125: Please state if you used flat-bottomed or round-bottomed microplate. Wording has been revised. The microplates are described by the manufacturer as U-bottomed (line 133).
  • Line 147: Did you mean "as described above"? Wording has been revised as suggested (line 155).
  • Line 151: Instead of "create" (meaning make something from nothing) I would suggest to write "make". Wording has been revised as suggested (line 163).
  • Line 162: "two-fold dilutions" (instead of "doubling dilutions"). Wording has been revised as suggested (line 170).
  • Line 172: What do you mean with "sighting assays"? Please describe better in the text. This section has been extensively revised and the reference to “sighting assays” has been removed.
  • Line 175: add "were" before "added". Wording has been revised as suggested.
  • Line 177: Please define the composition of OPKA buffer. (In line 184 – the authors used the concept "assay buffer"). Be consequent. Assay buffer has been described in methods (line 172). We have changed mentions of OPKA buffer to assay buffer for consistency.
  • Line 186: Please describe the composition of the controls.  It is not clearly presented. The minus sign is mistakenly taken as a hyphen. Please write in full name (minus or without). Revised in manuscript as suggested (line 180, 197).
  • Line 187: Define COH. Abbreviation defined in methods (line 102).
  • The method section should be written in a more scientific way. Very clumsy and cumbersome. We agree this section required revision and it has been extensively revised.
  • The calculations should be written in equations. We think that the revised text makes the calculations clear.
  • How is the survival curve plotted. Please write the formula. The description of titer determination (section 2.7) has been revised.
  • Line 213: What do you mean with " U/N"? This sentence has been deleted from the revised text.
  • Line 216: Who is GSK? GSK is a global pharmaceutical company formerly known as GlaxoSmithKline, now known as GSK. This has been revised to GSK plc (line 215).
  • Line 218: Do you mean 5-fold dilution? Wording has been revised as suggested (line 217).
  • Figure 1: In the figure: "No protein mAb binding" – Do you mean "no mAb binding?" Sorry for the confusion. We have revised wording to “No binding of mAb specific to alpha-like proteins”. The reviewer has mistaken this as the capsular polysaccharide. So how were these serotypes defined if the serotype-specific antibodies didn't bind to the bacteria? This comment is obsolete following clarification and revision to previous wording.
  • Line 216: It means that the assay only works with specific strains? Meaning a limitation of the assay? The assay works on all strains as shown in figure 2, but some variability exists within each serotype. The aim was to select one strain to represent each serotype. This development process is described in the main text of the manuscript and method sections.
  • What could be the reason for the phenomenon of outliers? We believe that differences in capsule expression may be one possible reason for the phenomenon of outlier strains and this is the subject of a follow-on study.
  • Line 273: " opsonophagocytic". The wording has been revised as suggested (line 283).
  • Are the representative bacteria available for the general audience? Yes, they are available for purchase at the National Collection of Type Cultures (NCTC, UK). An extra sentence has been added for clarification (lines 273-274).
  • Line 288: please reformulate – you can not say "facilitated by" – the dHL60 is the phagocytic cells, not the facilitator. The wording has been revised as suggested (line 302).
  • In Figure 4: correct to "dimethylformamide-differentiated HL60 cells". Edits have been made to figure 4.
  • How do you know if the HL60 cells are completely differentiated? Extra information has been added to section 2.4 of methods (lines 156-160).

We hope that you will find the revisions to the manuscript suitable and thank you for the helpful feedback